# Long-Term Consequences of COVID-19 Lockdown in Neovascular AMD Patients in Spain: Structural and Functional Outcomes after 1 Year of Standard Follow-Up and Treatment

**DOI:** 10.3390/jcm11175063

**Published:** 2022-08-28

**Authors:** Daniela Rego-Lorca, Alicia Valverde-Megías, José Ignacio Fernández-Vigo, Carlos Oribio-Quinto, Antonio Murciano-Cespedosa, Julia Sánchez-Quirós, Juan Donate-López, Julián García-Feijóo

**Affiliations:** 1Department of Ophthalmology, San Carlos Clinical Hospital, Instituto de Investigación Sanitaria del Hospital Clínico San Carlos (IdISSC), 28040 Madrid, Spain; 2Centro Internacional de Oftalmología Avanzada, 28010 Madrid, Spain; 3Department of Biodiversity, Ecology & Evolution, Faculty of Biology, Complutense University, 28040 Madrid, Spain; 4Modeling, Data Analysis and Computational Tools for Biology Research Group, Complutense University, 28040 Madrid, Spain; 5Neurocomputing and Neurorobotics Research Group, Complutense University, 28037 Madrid, Spain; 6Brain Plasticity Group, Instituto de Investigación Sanitaria Hospital Clínico San Carlos (IdISSC), 28040 Madrid, Spain

**Keywords:** COVID-19, pandemic, lockdown effects, nAMD, anti-VEGF

## Abstract

Consequences of the COVID-19 pandemic on medical care have been extensively analyzed. Specifically, in ophthalmology practice, patients suffering age-related macular degeneration (AMD) represent one of the most affected subgroups. After reporting the acute consequences of treatment suspension in neovascular AMD, we have now evaluated these same 242 patients (270 eyes) to assess if prior functional and anatomical situations can be restored after twelve months of regular follow-up and treatment. We compared data from visits before COVID-19 outbreak and the first visit after lockdown with data obtained in subsequent visits, until one year of follow-up was achieved. For each patient, rate of visual loss per year before COVID-19 pandemic, considered “natural history of treated AMD”, was calculated. This rate of visual loss significantly increased during the lockdown period and now, after twelve months of regular follow-up, is still higher than before COVID outbreak (3.1 vs. 1.6 ETDRS letters/year, *p* < 0.01). Percentage of OCT images showing active disease is now lower than before the lockdown period (51% vs. 65.3%, *p* = 0.0017). Although anatomic deterioration, regarding signs of active disease, can be apparently fully restored, our results suggest that functional consequences of temporary anti-VEGF treatment suspension are not entirely reversible after 12 months of treatment, as BCVA remains lower and visual loss rate is still higher than before the COVID-19 pandemic.

## 1. Introduction

Severe acute respiratory syndrome coronavirus type 2 (SARS-CoV-2) pandemic has had countless health and economic consequences worldwide. In an attempt to reduce the spread of COVID-19 and to optimize limited resources, outpatient visits were significantly reduced for months. In particular, ophthalmology consultations suffered one of the greatest decreases in patients’ visits [1]. More specifically, retina specialists had a particular conflict, as elderly patients, who had the greatest need for intravitreal injections to slow down visual impairment, are also those with the highest mortality and morbidity from SARS-CoV-2 infection [2]. Among these patients, age-related macular degeneration (AMD) was by far the most frequent pathology.

AMD is the leading cause of visual loss among patients older than 50 years in Western Europe [3]. Along with population aging, prevalence of AMD is expected to rise throughout the next decades. Among patients with neovascular AMD (nAMD), visual impairment occurs secondary to the ingrowth of macular neovascularization (MNV) within the retina, leading to bleeding and exudation [3]. Although only 20% of AMD patients develop nAMD, this form of the disease is responsible for around 90% of central vision loss associated with AMD [4]. Until the 21st century, there was no effective treatment. Recently, with anti-VEGF agents, the course of nAMD has dramatically changed.

SARS-CoV-2 pandemic reached Spain in January 2020. On 14 March 2020 a strict national lockdown was imposed in Spain, which lasted for more than 3 months, until 21 June. Fear of COVID-19 and difficult access to hospitals significantly reduced follow-up visits and treatment of nAMD patients during lockdown and for a few months thereafter. This unfortunate situation gave us the opportunity to study the anatomic and functional consequences of temporary anti-VEGF treatment suspension in nAMD patients.

We already reported the acute consequences of this treatment suspension in a previous study evaluating 242 patients whose follow-up and treatment was delayed, due to COVID-19 pandemic, for at least three months [5]. Now, one year later, we describe the functional and structural situation of these same patients, trying to assess whether ophthalmological consequences of COVID-19 lockdown can be fully or partially restored after twelve months of regular follow-up and treatment.

## 2. Materials and Methods

### 2.1. Study Participants

This study is a continuation of our previous research [5], now evaluating the functional and anatomic situation after 1 year of standard treatment (following pro re nata, fixed, or treat-and-extend regimen, as exposed in Table 1).

In this consecutive observational case series, all patients from San Carlos Clinical Hospital (Madrid, Spain) who were following anti-VEGF treatment for exudative AMD the year prior to COVID-19 lockdown were included. The study adhered to the 1964 Helsinki Declaration and was approved by San Carlos Clinical Hospital Ethics Committee. Written informed consent to use their medical information in the study analysis was routinely provided by all the patients.

Inclusion criteria for this study were: (i) neovascular AMD diagnosis, (ii) resuming of follow-up after the 14th of March, (iii) a period of at least twelve weeks between the visits before and after lockdown onset to prevent confounding factors with more usual delays in clinical practice, and (iv) having records of complete ophthalmological examinations carried out during the immediate two visits before lockdown (named COVID-1 and COVID-2), the visit after the lockdown onset (COVID 0), and subsequent visits after that (COVID+1, COVID+2…) until 1 year of follow-up was achieved (COVID/last).

Exclusion criteria were: (i) macular neovascularization (MNV) due to causes other than AMD, (ii) patients returning to hospital for visit COVID 0 after January 2021, (iii) visual acuity (VA) of counting fingers or less before lockdown, and (iv) loading dose not completed before lockdown.

Out of the 270 eyes evaluated in our first study, 23 patients (25 eyes) ceased follow-up during the year after COVID 0 visit, so they were excluded from the present study. Reasons were death (*n* = 12), referral hospital change (*n* = 3), visit discontinuation due to major health problems (*n* = 3), and unknown causes (*n* = 5).

At all visits, patients received a complete ophthalmological examination including measurement of best corrected visual acuity (BCVA) using an Early Treatment Diabetic Retinopathy Study (ETDRS) letters chart, dilated ophthalmoscopy, slit lamp biomicroscopy, dilated fundus examination, structural optical coherence tomography (OCT) imaging, and, when prescribed, anti-VEGF intravitreal injection. For each patient, information about diagnosis (date, type of neovascularization and baseline BCVA), number of intravitreal injections one and two years before lockdown, anti-VEGF used, and protocol applied was extracted from medical records. For the analysis, and to account for the variability inherent to this disease, we used data from the immediate two visits before lockdown (COVID-2 and COVID-1), which were on average 59 days apart, then the visit after the onset of the lockdown (COVID 0) and subsequent visits (COVID+1, COVID+2…) until 1 year of follow-up was achieved (COVID/last).

### 2.2. OCT Imaging

Structural OCT imaging was performed with the Heidelberg Spectralis OCT device (Heidelberg Engineering, Heidelberg, Germany). Each set of scans included 25 horizontal B-scans, centered on the fovea, with a minimum strength signal of 25 as recommended [6]. Exudative disease activity was assessed as active/inactive MNV and presence of subretinal fluid (SRF) and intraretinal fluid (IRF). Macular cystoid edema was recorded and central retinal thickness (CRT) was measured. Structural OCT images were reviewed by two independent and experienced readers (A.V.M. and D.R.L.). A third reader was advised in case of disagreement in the scan assessment (JI.F.-V.).

### 2.3. Statistical Analysis

Statistical calculations were performed using Statgraphics Centurion (version 19, Statgraphics Technologies, Inc., The Plains, VA, USA). Normality of variables was assessed with Shapiro–Wilk test. Statistical significance of the differences for binomial variables was assessed using proportion comparisons with normal approximation. Mann–Whitney test was used to compare BCVA and CRT, respectively, at different visits (COVID-1, COVID 0, COVID/last…). A *p* value < 0.05 was considered statistically significant.

## 3. Results

### 3.1. Characteristics of the Eyes Included in the Analysis

From the 242 nAMD patients (270 eyes) who suffered anti-VEGF treatment delay of three or more months included in our previous study, 219 patients (245 eyes) completed one year of follow-up since regular visits were resumed after lockdown. Among them, 171 (69.8%) were women.

The baseline demographic data of the 245 eyes included in the present analysis are shown in Table 1. The year of diagnosis of exudative AMD ranged from 2006 to 2020. All types of MNV were represented, although the most common was type 1 (65.7%), and the least common was what we used to call polypoidal choroidal vasculopathy, now formally known as aneurysmal type 1 neovascularization (2%). Anti-VEGF agent used included ranibizumab (38.8%), aflibercept (36.3%), and bevacizumab (24.9%). The number of injections in the year preceding the lockdown was 5.4 ± 1.8 (range 1–11). Real delay (101.4 ± 56.6 days) was calculated as time from their planned visit to the date the patients resumed their follow-up.

### 3.2. Functional and Anatomic Outcomes

#### 3.2.1. Visual Acuity

Mean BCVA before lockdown, at COVID-1 visit, was 60.5 ETDRS letters. BCVA at COVID-2, COVID-1, COVID 0, COVID+1, and COVID/last visits is shown in Table 2. There were no statistically significant differences between BCVA in COVID-2 and COVID-1. VA loss assessed after the lockdown period was 3.9 ETDRS letters (*p* = 0.028), as mean BCVA at COVID 0 visit had worsened to 56.6 ETDRS letters. After 1 year of regular follow-up and treatment, at visit COVID/last, BCVA was 53.3 ETDRS letters, even lower than after COVID lockdown. However, differences in VA among the visits conducted during the year after restarting regular treatment are not significantly different (*p* > 0.6).

For each patient we calculated, using medical records starting from VA after loading dose in the year of diagnosis, the rate of VA loss per year before COVID-19 outbreak. This concept was considered visual loss not attributable to treatment delay, but to “natural history of treated nAMD”. Then, in our cohort, we calculated rate of VA loss throughout the year prior to COVID lockdown which was 1.6 ETDRS letters/year. Mean BCVA before COVID lockdown was, as mentioned, 60.5 ETDRS letters. When follow-up was resumed, after an average time of 171 days (mean time between COVID-1 and COVID 0), BCVA was 56.6 ETDRS letters. This 3.9-letter loss is consequent to a VA loss rate during this time, between COVID-1 and COVID 0, of 8.6 ETDRS letters/year. As the expected VA loss attributable to “natural history of treated nAMD” during this same period would be 0.6 letters (1.6 letters/year VA loss for 171 days), we presume that 3.3 ETDRS letters loss (from the actual 3.9 ETDRS letters decrease) is attributable to COVID-19-induced delay.

Twelve months after resuming follow-up (379.5 days, mean time between COVID 0 and COVID/last), VA loss rate is still higher than before lockdown (3.1 ETDRS letters/year vs. 1.6 ETDRS letters/year, *p* < 0.01). Mean VA decreased from 56.6 letters at COVID 0 to 53.3 letters at COVID/last, a mean decrease of 3.3 letters according to mentioned VA loss rate of 3.1 letters/year. Finally, in a complete overview, since lockdown was imposed until 12 months after resuming follow-up (550.5 days, mean time between COVID-1 and COVID/last), estimated VA loss related to “natural history of treated nAMD” would be 2.5 ETDRS letters (1.6 letters/year VA loss rate for 550.5 days). However, mean VA loss was 7.2 ETDRS letters (from 60.5 letters at COVID-1 to 53.3 letters at COVID/last). Thus, we presume that 4.7 ETDRS letters lost (from the 7.2 letters decrease) since COVID-19 outbreak can be attributed to this treatment suspension. Rates of VA loss for each period and expected and real BCVA are shown in Figure 1.

#### 3.2.2. Disease Activity Assessed by OCT

Structural parameters and tomographic features determined by OCT imaging are shown in Table 3 and Figure 2.

Active disease was defined as presence of SRF, IRF, or both. OCT evaluation revealed active disease in 65.3% of eyes in COVID-1. After the lockdown period, percentage of OCT images classified as showing active disease increased to 79.6%. However, at COVID/last visit, following 1 year of standard treatment, there were significantly less OCT images with evidence of active disease as compared with both after and before the lockdown period (*p* < 0.001 and *p* = 0.0017, respectively). Percentage of eyes showing disease activity in OCT images at COVID/last was 51%, less than before the delayed treatment period.

Concerning fluid distribution in OCT, before COVID lockdown, SRF was present in 25.7% of eyes, IRF in 21.6%, and 14.7% of OCT images showed both SRF and IRF; 3.3% of eyes presented cystoid macular edema (CME). After the period of treatment delay, the group of patients with presence of both SRF and IRF suffered the greatest increase, from 14.7% at COVID-1 to 24.5% at COVID 0. The percentage of these eyes, showing both SRF and IRF, was significantly reduced after 1 year of regular follow-up and treatment (10.6% at COVID/last vs. 24.5% at COVID 0, *p* < 0.001), returning to values comparable to those observed before lockdown (10.6% at COVID/last vs. 14.7% at COVID-1, *p* = 0.17). At this same point, following 1 year of standard treatment, the percentage of eyes showing SRF in OCT images significantly decreased, reaching levels below those observed before COVID-19 outbreak (14.7% at COVID/last vs. 25.7% at COVID-1, *p* = 0.024). Comparison of eyes presenting IRF at COVID/last and COVID-1 visits did not yield statistically significant differences (25.7% at COVID/last vs. 21.6% at COVID-1, *p* = 0.289).

CRT was also measured in each OCT image. Before the COVID-19 pandemic, mean CRT was 303.8 ± 162.2 µm. Although following the period of treatment delay a thicker mean CRT was observed, after 1 year of standard treatment, CRT decreased, recovering values similar to those observed before COVID-19 outbreak (293.3 µm at COVID/last vs. 303.8 µm at COVID-1, *p* = 0.117). This reduction was already observed in COVID+1 visit, the first visit after the one resuming follow-up, when mean CRT was 290.2 ± 170.5 µm.

#### 3.2.3. Intravitreal Anti-VEGF Treatment

The year prior to COVID-19 outbreak mean number of anti-VEGF intravitreal injections was 5.4 ± 1.8. During the year between COVID 0 and COVID/last visits, the mean number of injections increased to 6 ± 2.7. In the first trimester after lockdown, the number of intravitreal injections was significantly higher than in the other quarters of the year (2.1 vs. 1.4; 1.3, and 1.3; at first, second, third and fourth trimester, respectively; *p* < 0.05).

## 4. Discussion

COVID-19 pandemic has had severe consequences regarding medical care. Among AMD patients, anatomic and functional outcomes after COVID-19 lockdown have been widely analyzed [7,8,9,10,11,12,13,14,15]. One year after pandemic outbreak and national lockdown, we are now able to assess long-term consequences of anti-VEGF treatment delay studying the anatomic and functional situation of these same patients after one year of regular follow-up and treatment.

In our study, after a period of intensive treatment, especially during the first trimester after follow-up recovery, we achieved percentages of exudative disease on OCT images that are even lower than those we had before lockdown. Unfortunately, functional outcomes do not match the anatomic results, as VA remains lower than before COVID-19 pandemic. Although the percentage of OCT images with evidence of active disease and the average central retinal thickness are below those described before lockdown, VA has not reached pre-confinement values, and the rate of VA loss, while lower than during the lockdown period, remains higher than before COVID outbreak.

As mentioned above, we only included patients who suffered a delay of more than 12 weeks, trying to prevent confounding factors with more usual delays in clinical practice. This decision was reinforced by the outcomes reported by Douglas et al. [16]. These authors compared VA changes between patients with less than 6 weeks delay and patients with more than 12 weeks delay, finding significantly higher VA loss among patients with more than 12 weeks delay.

Our results are similar to those found by Stattin et al. [17] in the only other published study reporting outcomes at one-year follow-up after resuming normal practice. In this retrospective study, 98 nAMD patients who suffered an average treatment deferral of 61.1 ± 15 days (mean injection interval was extended from planned 56.5 ± 27.7 days to 117.6 ± 31.4 days) were evaluated. Mean VA before lockdown was higher in their sample than in ours (67.2 vs. 60.5), but this can be explained by the fact that they excluded patients with VA of less than 40 ETDRS letters. In our study, only patients with visual acuity of counting fingers or less were excluded. Stattin et al. report that the number of visits or the number of intravitreal injections along the year after follow-up was resumed have no influence on VA change. However, they do find that the length of the treatment delay interval significantly affected VA loss after one year (regression estimate [RE] −0.034; 95% CI −0.033 to −0.013; *p* < 0.0001). This finding could explain the fact that VA loss after lockdown period is slightly higher in our study than that reported by Stattin et al. (3.2 vs. 2.2 ETDRS letters), as we only included patients with a period of at least 12 weeks between visits before and after lockdown onset, and mean delay in the Stattin et al. cohort is 9 weeks. Despite these minor differences, final VA loss is comparable between both studies, with a difference of less than 1 ETDRS letter. Stattin et al. found a VA loss after one-year follow-up of 4.1 ETDRS letters, as compared with VA before lockdown (*p* < 0.0001). In our study, final VA loss attributable to COVID-19 lockdown was 4.7 ETDRS letters. Unfortunately, in their study, Stattin et al, only evaluated VA changes, so anatomic outcomes could not be compared.

Similar results are found in studies reporting outcomes after 6-month follow-up [18,19,20,21]. All of them describe baseline VA comparable to that in our cohort, and analogous to our results, they report that mean final VA that does not return to pre-pandemic values. Arruabarrena et al. [21], similar to our previous findings, report that 48.15% of their patients maintain VA with changes of less than ± 5 ETDRS letters after 6-month follow-up.

Regarding anatomic outcomes, Rush et al. [18] found that, although central retinal thickness (CRT) decreased following an initial increase after the lockdown period, 6 months later it remained higher than before COVID-19 outbreak. Another study assessing anatomic results is that of Zarranz et al. [20], a multicenter study which describes, in Spain, a 3.1% higher proportion of active disease in OCT images 6 months after the beginning of COVID lockdown. Yeter et al. [22], after a mean follow-up period of 3.5 ± 1 months subsequent to COVID lockdown, found that CRT decreased almost to pre-pandemic levels and that OCT findings (SRF, IRF, subretinal hyperreflective material (SRHM)) regressed to the frequency observed in pre-lockdown visits (80% vs. 35%, *p* < 0.001; 51% vs. 29%, *p* = 0.022; and 31% vs. 11%, *p* = 0.01, for SRF, IRF, and SRHM at first visit after restrictions vs. last visit after restrictions, respectively). In our study, after a 12-month follow-up period, we observed a decrease in the percentage of OCT images with active disease and in CRT, reaching values below those observed before COVID-19 pandemic. Nevertheless, these differences may be justified by their shorter follow-up (Rush et al.: 6 months after resuming follow-up; Zarranz et al.: 6 months since lockdown beginning; Yeter et al.: 3.5 months after resuming follow-up), which might have prevented them from detecting later improvements. Additionally, both Zarranz et al. and Rush et al. refer a smaller number of intravitreal injections after lockdown, which could also explain differences with our results, as in our study, parallel to what Stattin et al. describe, we found a higher frequency of intravitreal injections following the period of treatment delay.

Although the situation triggered by the SARS-CoV-2 pandemic has been unique, there were already some studies, conducted before COVID-19 outbreak, emphasizing the negative impact of delaying anti-VEGF treatment of active nAMD. In this regard, Chong Teo et al. conducted the RAMPS study, where 286 patients were divided into timely or delayed re-treatment groups. Delayed re-treatment was defined as eyes that did not receive anti-VEGF treatment on twot or more visits despite diagnosis of active disease. The authors report that, although the number of intravitreal injections over 12 months was not statistically different, timely treated patients had greater VA gains at 12 months than the delayed treatment group (6.4 vs. 1.2 ETDRS letters, *p* = 0.04). Anatomic results were analogous, as timely patients had greater reduction on CRT at 12 months than patients in the delayed treatment group (135 µm vs. 87.8 µm, *p* = 0.04). Moreover, Chong Teo et al. report that longer delay between detection of active disease and re-treatment was associated with poorer VA (*p* = 0.03), as VA decreased 0.02 ETDRS letters per day of treatment delay. These results are in line with our findings, as we also found a decrease of 0.02 ETDRS letters per day (8.6 ETDRS letters/year) during the treatment delay period. Consistent results were also reported by Muether et al. in two different studies [23,24], concluding that deferral between indication to treat and treatment administration could be responsible for a deterioration of VA that may not be completely reversible by restarting scheduled anti-VEGF treatment. This same idea is concluded comparing MARINA [25] and ANCHOR [26] clinical trials with PIER [27] clinical trial, as monthly anti-VEGF injections showed better functional results than quarterly administration, suggesting that recurrence of active disease between injections may lead to permanent damage.

The unfortunate situation suffered by these patients has taught us that temporary suspension of anti-VEGF treatment in nAMD has functional consequences that are not entirely reversible after 12 months of standard treatment. Although anatomic deterioration can be apparently fully restored with regular intravitreal treatment, the damage already induced to the retina seems to prevent VA from reaching previous levels. This understanding might help us and our patients taking informed decisions if another emergency impelling outpatients visits suspension is to come.

## 5. Conclusions

AMD patients who suffered a delay of 3 or more months in anti-VEGF treatment, although having achieved anatomic situations apparently better than those observed before COVID-19 outbreak after 1 year of regular follow-up and treatment, maintain VA lower than before lockdown and are still losing vision at a higher rate than before the period of treatment delay.

## Figures and Tables

**Figure 1 jcm-11-05063-f001:**
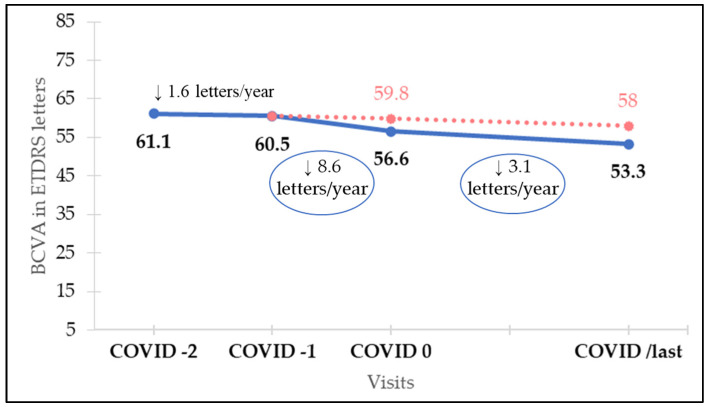
Rates of visual loss for each period and real (blue line) and expected (red dotted line) best corrected visual acuity (BCVA).

**Figure 2 jcm-11-05063-f002:**
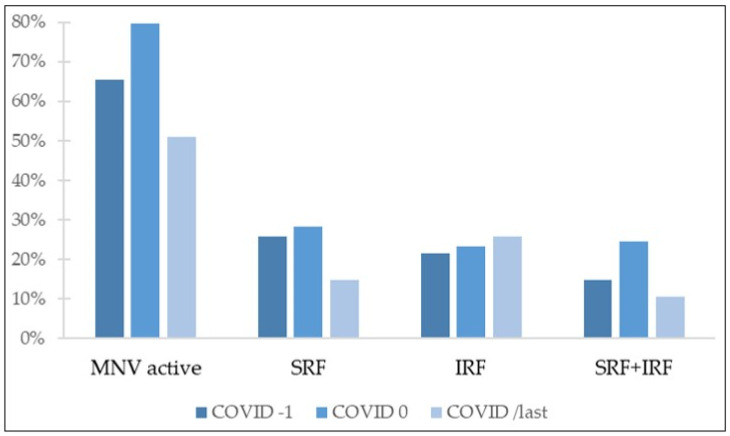
Percentage of eyes with different tomographic features. MNV: macular neovascularization; SRF: subretinal fluid; IRF: intraretinal fluid.

**Table 1 jcm-11-05063-t001:** Demographic variables assessed in AMD patients included in the study.

Age (years), mean (SD)	82.5 (6.3)
Patients with both eyes eligible (%)	26 (10.6%)
Years since MNV diagnosis, mean (SD; range)	4.9 (3.2; 1–14.3)
MNV type, *n* (%)	
Type 1	161 (65.7%)
Type 2	50 (20.4%)
Type 3	29 (11.8%)
AT-1	5 (2%)
Anti-VEGF used, *n* (%)	
Ranibizumab	95 (38.8%)
Aflibercept	89 (36.3%)
Bevacizumab	61 (24.9%)
Regimen	
Pro re nata	102 (41.6%)
Treat-and-extend	37 (15.1%)
Fixed	106 (43.3%)
Anti-VEGF injections, mean (SD; range)	5.4 (1.8; 1–11)
Delay in follow-up/treatment (days), mean (SD; range)	101.4 (56.6; 28–298)

SD: standard deviation; MNV: macular neovascularization.

**Table 2 jcm-11-05063-t002:** BCVA obtained in each visit was compared with BCVA before (COVID-1) and after (COVID 0) lockdown.

	ETDRS Letters, Mean (SD; Range)	vs. COVID 0 *p* Value	vs. COVID-1 *p* Value
COVID-2	61.1 (18.7; 5–91)	0.006	0.58
COVID-1	60.5 (18.5; 5–85)	0.028	-
COVID 0	56.6 (20.4; 5–90)	-	0.028
COVID+1	55.8 (20.6; 5–90)	0.57	0.0075
COVID/last	53.3 (22; 5–90)	0.11	<0.001

ETDRS: Early Treatment Diabetic Retinopathy Study. SD: standard deviation.

**Table 3 jcm-11-05063-t003:** Structural parameters assessed by OCT at different visits.

	Active MNV *n* (%)	SRF *n* (%)	IRF *n* (%)	SRF and IRF *n* (%)	CME *n* (%)	CRT Mean ± SD
COVID-1	160 (65.3%)	63 (25.7%)	53 (21.6%)	36 (14.7%)	8 (3.3%)	303.8 ± 162.2
COVID 0	195 (79.6%)	69 (28.2%)	57 (23.3%)	60 (24.5%)	8 (3.3%)	343.9 ± 186.6
COVID/last	125 (51%)	36 (14.7%)	63 (25.7%)	26 (10.6%)	0 (0%)	293.4 ± 211.8
COVID/last vs. COVID 0 *p* value	<0.001	<0.001	0.532	<0.001	0.0043	<0.001
COVID/last vs. COVID-1 *p* value	0.0017	<0.001	0.289	0.17	0.0043	0.117

MNV: macular neovascularization; SRF: subretinal fluid; IRF: intraretinal fluid; CME: cystoid macular edema; CRT: central retinal thickness.

## Data Availability

Data used to support the findings presented in this study are available on request from the corresponding author.

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
