# Peer review of "Long-Term Consequences of COVID-19 Lockdown in Neovascular AMD Patients in Spain: Structural and Functional Outcomes after 1 Year of Standard Follow-Up and Treatment"

_jcm, 2022, doi:10.3390/jcm11175063_

Round 1

Reviewer 1 Report

I would like to congratulate with the Authors for their interesting work. The manuscript has a good readability and try to evaluate the consequences of delayed treatment caused to COVID -19 lockdown.

Introduction is well written and provide sufficient background and aims of the study

Methods: the authors use the data of the two visits before lockdown (Covid -2 and Covid -1), but they did not mention how many weeks they date back to.

Results: the authors state that VA loss prior COVID lockdown was 1.6 ETDRS letters/years. It is not immediately clear how they calculated this data. It also applies to “natural history of treated nAMD” during lockdown (0.6 letters) and “estimated VA loss of 2.5 ETDRS letters (lines 169-170). They should better explain how they calculated this data.  

Line 202: extra space between “observed” and “before”

Despite the authors stated that they used “Generalized linear models (GLMs) were used to assess the relationship between changes in BCVA (dependent variable) and other clinical features and demographics (independent variables). Mann Whitney test was used to assess the relationship between BCVA and CRT. Post hoc analyses were performed with Wald tests.”(lines 120-124)  I was not able to find these analysis in results. 

Author Response

Thank you very much for your comments. They are interesting and have helped to improve our work.

Covid-2 and Covid-1 were on average 59 days apart. This is now mentioned in the methods section.

We used VA from medical records starting at first visit after loading dose the year of diagnosis to calculate rate of VA loss/year. This was considered “natural history of treated nAMD”. Then, based on the data for the year before COVID outbreak, average rate of VA loss was 1.6 letters/year. According to this rate of VA loss prior to COVID outbreak, applying these data to each patient, estimated VA loss during the delay period would be on average 2.5 EDTRS letters. We have updated this paragraph trying to make it clearer.

In the “statistical analysis” section there are some major mistakes. We feel very sorry for the inconvenience. In our first article (PMID: 34441845) we did use generalized linear models to assess the relationship between changes in BCVA and other clinical features and post hoc analyses with Wald tests were performed, but not again in this article. This section has been corrected in this new version. Again, we apologize for the inconvenience caused.

Reviewer 2 Report

Dear Author(s),

Thanks for your submission on the JCM. Your article is an extensive and precise report of the nAMD behaviour in a large cohort of patients after the Spain lockdown and the "restart" of a standard therapy. The results obtained are in line with many trials and reports on the use of anti-VEGF drugs, which you in part appropriately cited. In my opinion this is a high-quality article and needs only a minor language revision. I would suggest to use the term MNV (macular neovascularization) as the new standard in terminology throughout the text. Did you observe an increase in the fibrotic tissue deposition or ellipsoid damage in patients with delayed Intravitreali therapy? It would be useful to understand if other OCT parameters, other than those indicating the activity of the disease, are connected with a slight poor prognosis in this type of cohort.

Author Response

Thank you very much for your comments. They are interesting and have helped to improve our work.

We have now used the term MNV (macular neovascularization), as you pointed out it is the new standard terminology.

We only evaluated the presence of OCT parameters indicating activity of the disease. However, as you propose, it would be interesting in future studies to reevaluate the OCT images and assess if any other parameters are related to visual prognosis.

Reviewer 3 Report

Dear authors:

The work is exciting and allows to evaluate the impairment in visual acuity of patients who suffer delays in their treatment with anti-VEGF. 

1. The aim of the work is not clear enough: it is to evaluate the effect of delay in anti-VEGF treatment of patients with AMD but is it different from the delay in anti-VEGF therapies due to the pandemic of COVID?

2. The authors propose in the statistical analysis to perform a multivariate linear regression model. Still, the results do not appear except for a bivariate regression between treatment delay and visual acuity. It would be interesting to establish such a model to assess the importance of the different variables in the final result. 

3. Although there are multiple studies relating the results of the three antiVEGFs used in the study, the effect of the three antiVEGFs should be studied separately, how the delay affects each treatment and how they influence the final result.

Author Response

Thank you very much for your comments. They are interesting and have helped to improve our work.

1. In the last paragraph of the “introduction” section we explain: “we already reported the acute consequences of this treatment suspension in a previous study evaluating 242 patients whose follow-up and treatment was delayed, due to COVID-19 pandemic, for at least three months [5]. Now, one year later, we describe the functional and structural situation of these same patients, trying to assess whether ophthalmological consequences of COVID-19 lockdown can be fully or partially restored after twelve months of regular follow-up and treatment”. I am not sure I have understood the question correctly, if you have any doubt please do not hesitate to let us know.

2. In the “statistical analysis” section there are some major mistakes. We feel very sorry for the inconvenience. In our first article (PMID: 34441845) we did use generalized linear models to assess the relationship between changes in BCVA and other clinical features and post hoc analyses with Wald tests were performed, but not again in this article. This section has been corrected in this new version. Again, we apologize for the inconvenience caused.

3. In a first analysis carried out and reported in our first article (PMID: 34441845) we found that the type of anti-VEGF used was not related to the functional outcomes. However, as you propose, it could be interesting to evaluated more extensively in future studies differences in this regard.

Reviewer 4 Report

This is an interesting and overall well-designed and written study. Here are a few comments and suggestions.

With regards to the inclusion/exclusion criteria, did any of the patients develop CNV at any follow up? Please update the methods section accordingly and comment on how you handled such cases if any.

In addition, in the OCT imaging section please mention if a third reader was advised in case of disagreement in the scan assessment.

I would suggest to include data regarding the gender (%).

It would also be interesting to see the outcomes based on the age interquartile range. You can review and reference a recent study published also by JCM where the short- as well the long-term visual outcomes of patients with neovascular AMD were investigated PMID: 35456189. In this study the authors also investigated whether there is a correlation between final VA and duration of delay. 

An additional study where long-term outcomes were investigated is this: PMID: 35596203.  

In some patients, data from both eyes has been included. Please explain how you treated the data from a statistical point of view and whether you further ran a subanalysis for both structural and functional parameters.

Were data of fellow eyes collected? If yes, it would be interesting to see the results. Were patients with VA of the fellow eye of counting fingers or less included in the analysis?

I would be happy to review the revised version of the manuscript.

Author Response

Thank you very much for your comments. They are interesting and have helped to improve our work.

All patients included had already been diagnosed with CNV and treated with at least three antiVEGF intravitreal injections. We have now specified that among the inclusion criteria.

A third reader was advised in case of disagreement in the scan assessment. This is now stated in the “methods section”.

We have now included in the “results section” data regarding the gender. 60.8% of our patients were women.

Both studies provide some interesting data. In the first one (PMID: 35456189) the authors compare VA changes between patients with <6 weeks delay and patients with >12 weeks delay, finding statistically significant differences. In line with these findings, we believe that patients with delays <12 weeks should be excluded to prevent confounding factors with more usual delays in clinical practice. We have now included this article among our bibliography.

We included both eyes of patients with bilateral nAMD, treating each eye as independent data. We initially approached it this way because we believe that, when patients suffer this disease bilaterally, usually each eye is diagnosed at a different stage of the disease. So, frequently, when both eyes are involved, one of them is already at an advanced stage with considerably lower visual acuity as compared to the fellow eye. Therefore, we thought that each eye of a same patient, being at a different stage of the disease and often following different frequency of intravitreal treatment, might respond differently to the treatment delay that we are evaluating here. Nevertheless, we evaluated possible correlations between both eyes of these patients regarding VA and CRT changes since COVID-1 visit until COVID 0 and COVID/last, finding in all cases p values > 0.05.

No, we did not collect data of the fellow eyes. However, as you propose, further studies evaluating data of fellow eyes would be very interesting. While we did not collect data from the fellow eyes, VA of the fellow eye was not among exclusion criteria.

Round 2

Reviewer 1 Report

I appreciate the authors' effort to improve the manuscript and I would like to thank them for that. However, there is still something unclear. 

If the authors calculated VA loss throughout the year prior to COVID lockdown, it means that they probably used data other than Covid -2 and COVID -1, since they were on average 59 days apart. This should be mentioned in the methods.  furthermore, if the natural history VA loss was 1.6 EDTRS letters, and the lockdown period was 3 months, VA loss during the lockdown period due to the natural history of the disease was expected to be 0.39 letters. How did they obtain 0.6 letters? 

If the mean VA was 56.6 at COVD 0 and 53.3 at Covid/last, VA loss in the year following lockdown should be 3.3 ETDRS letters/year and not 3.1 letters/year as mentioned in the text and in figure 1.

The authors could update the row data as a supplementary file or in a repository. 

Similarly, it is not clear how the Authors calculated 8.6 letters/year reduction from Covid -1 to Covid 0.

The relationship between clinical features and demographics could provide further interesting information.  

Author Response

Thank you again for your comments. We agree that the section regarding VA loss might not be clear enough. We will try to clarify this to you and in the manuscript.

Responding to your questions:

If the natural history VA loss was 1.6 EDTRS letters, and the lockdown period was 3 months, VA loss during the lockdown period due to the natural history of the disease was expected to be 0.39 letters. How did they obtain 0.6?

Natura history VA loss was 1.6 ETDRS letters, but the lockdown period was not 3 months. One of the inclusion criteria was a period of at least 3 months between the visits before and after lockdown, but our actual average lockdown period (or time between COVID -1 and COVID 0) was 171 days. Thus, with a rate of VA loss of 1.6 letters/year (0.004 letters/day) we obtain the VA loss during lockdown period of 0.6 (0.004*171=0.68). (0.6 instead of 0.7 is probably due to calculations made here using fewer decimal places)

If the mean VA was 56.6 at COVID 0 and 53.3 at COVID/last, VA loss in the year following lockdown should be 3.3 ETDRS letters/year and not 3.1 letters/year as mentioned in the text and in figure 1.

This is explained in a similar way to the above. 3.3 ETDRS letters/year would be the VA loss if the period between COVID 0 and COVID/last was exactly 365 days for every patient. However, this period was on average 379.5 days. Therefore, VA loss of 3.3 letters during 379.5 days leaves a VA loss rate of 0.0086 letters/day (3.3/379.5) or 3.1 letters/year.

Similarly, it is not clear how the Authors calculated 8.6 letters/year reduction from COVID -1 to COVID 0.

During the period between COVID -1 and COVID 0 (171 days, as mentioned before), a VA loss of 3.9 letters happened. Using these data, a VA loss rate of 0.023 letters/day or 8.4 letters/year is obtained (3.9/171). (8.4 instead of 8.6 is probably due to calculations made here using fewer decimal places)